# Experiences of the COVID-19 Lockdown and Telehealth in Aotearoa New Zealand: Lessons and Insights from Mental Health Clinicians

**DOI:** 10.3390/ijerph20064791

**Published:** 2023-03-08

**Authors:** Benjamin J. Werkmeister, Anne M. Haase, Theresa Fleming, Tara N. Officer

**Affiliations:** 1School of Health, Te Herenga Waka-Victoria University of Wellington, Wellington 6012, New Zealand; 2Te Whatu Ora-Health New Zealand, Psychological Medicine, Wellington 6021, New Zealand; 3Department of Psychological Medicine, University of Otago-Wellington, Wellington 6242, New Zealand; 4School of Nursing, Midwifery, and Health Practice, Te Herenga Waka-Victoria University of Wellington, Wellington 6021, New Zealand

**Keywords:** COVID-19, mental health, clinician, wellbeing, telehealth, interpretive description

## Abstract

Objective: The COVID-19 pandemic rapidly changed health service delivery and daily life. There is limited research exploring health professional experiences with these changes. This research explores mental health clinicians’ experiences over the first COVID-19 lockdown in New Zealand to inform future pandemic responses and improve usual business practices. Method: Thirty-three outpatient mental health clinicians in three Aotearoa New Zealand regions took part in semi-structured interviews. Interviews were analysed thematically applying an interpretive description methodology. Results: Three key themes emerged: (1) life in lockdown, (2) collegial support, and (3) maintaining well-being. Clinicians, fearful of contracting COVID-19, struggled to adapt to working from home while maintaining their well-being, due to a lack of resources, inadequate pandemic planning, and poor communication between management and clinicians. They were uncomfortable bringing clients notionally into their own homes, and found it difficult to separate home and work spheres. Māori clinicians reported feeling displaced from their clients and community. Conclusion: Rapid changes in service delivery negatively impacted clinician well-being. This impact is not lessened by a return to normal work conditions. Additional support is required to improve clinician work conditions and ensure adequate resourcing and supervision to enable clinicians to work effectively within a pandemic context.

## 1. Introduction

Aotearoa New Zealand’s (NZ) 2020 lockdown was amongst the most stringent globally [1]. On 21 March 2020, the NZ Government introduced a four-level alert system to manage COVID-19. The system controlled community transmission of COVID-19 through isolation. Alert levels 3 and 4 restricted travel to essential services, including supermarkets and medical care [2], these levels represented a ‘lockdown’ in NZ where people remained within their ‘bubbles’ (a metaphor for isolating in groups).

Lockdowns temporarily eliminated COVID-19 but had considerable negative effects on public well-being [3,4,5,6]. Every-Palmer, et al. [4] reported approximately 33% of participants experienced moderate to severe psychological distress, 17% moderate-to-severe anxiety, and 2% attempted suicide. Furthermore, employment losses, financial strain, and housing stress from COVID-19 lockdown restrictions contributed to the psychological toll on NZ’s population [7,8].

While population well-being research during the pandemic has grown, workforce well-being research has not seen the same expansion. There is increasing recognition of capacity limitations and associated mental health clinician burnout [9]. Internationally, research into clinicians’ pandemic experiences has focused on medical and nursing workforces, is largely survey-based, and limited studies have investigated other members of the mental health multidisciplinary team (MDT). Healthcare workers exposed to COVID-19 (either personally or through working with COVID-19-positive patients) had poorer mental health; burnout and anxiety were commonly reported [10,11]. The overwhelming demand for services may contribute to psychological trauma in these workers [12]. NZ research suggests disruption of personal life and isolation exacted emotional tolls on healthcare workers [13]. Greater in-depth understanding of mental health clinician wellbeing is necessary to refine clinician support systems, navigate future pandemic responses, and improve usual business practices.

In NZ, specialist mental health services care for the most at-risk clients; between 2011 and 2017, there was a 23% increase in client numbers [14]. Despite funding increases, these services are under-resourced, with long wait times, high thresholds for receiving care, and poor clinician retention and burnout [15,16,17,18,19]. Furthermore, approximately 50% of nurses and psychiatrists are over 50 years old, suggesting an impending increase in attrition in these groups [17]. Mental health clinicians comprise those from professions including; medicine (9.5%) (psychiatrist, registrar, and medical officer), nursing (45.2%) (registered and enrolled nurses and nurses with more advanced practice scopes), social work (6.7%), clinical psychology (6.1%), occupational therapy (3.9%), psychotherapy (0.3%), case management (16.9%), and cultural roles (1.2%) (including kaumātua (respected Māori elder) and cultural therapists) [17]. Outpatient teams are led by team leaders with one of the above roles who also coordinates service provision and resourcing with management.

Māori, New Zealand’s Indigenous population, comprise approximately 16% of New Zealand’s population but account for 28% of all mental health clients [14] and generally exhibit poorer mental health outcomes [16,20]. Māori identify that ‘generational deprivation’ and colonization have negatively impacted Māori mental health, and Western models of mental health care do not fit with Māori understandings of mental health [16]. Health services in New Zealand have obligations under Te Tiriti o Waitangi (the Treaty of Waitangi), to address the needs of Māori patients and reduce inequities experienced by this population. To address inequities in mental health service provision, the NZ Ministry of Health [21] and New Zealand Government [16] proposed that more Māori clinicians are required to provide services for Māori clients within Kaupapa Māori mental health services (mental health services that have underlying philosophies and practices reflecting Māori cultural values [21]). However, Māori clinicians are underrepresented in the NZ mental health workforce [17,22], with only 2% of senior doctors and 14% of all mental health practitioners identifying as Māori [17].

In the current austere mental health environment, the pandemic forced changes in service provision, including telehealth-based care (health services delivered via telecommunications), and dividing outpatient teams into sub-teams to maintain care continuity and limit viral transmission. While there is a growing body of research into clinicians experiences delivering telehealth-based mental health services (see for example, [19]), there is less research into the wider experiences of clinicians over the COVID-19 lockdown.

The present study investigates the experiences of clinicians from multiple clinical backgrounds working in outpatient mental health teams in three NZ regions. These regions serviced approximately 16,230 clients in 2020/2021, representing approximately 10% of all clients receiving publicly funded NZ specialist mental health services [23].

## 2. Materials and Methods

This study applied an interpretive description methodology [24] to develop a shared understanding of the challenges mental health clinicians faced during NZ’s first COVID-19 lockdown, and their experience adopting telehealth to provide mental health services. Interpretive description borrows and adapts aspects of ethnography, phenomenology, and grounded theory to explore clinically relevant problems in practice [24,25]. This methodology does not enforce the application of a singular rigid method or research design. Instead, the methodology acts as an “operating logic” [26] unfettered by theoretical approaches and consequently, able to be applied to qualitative and mixed methods research. This flexibility serves an important purpose wherein the researcher can draw information from multiple sources to then make cogent arguments linking research findings to clinical practice, as in this research.

Figure 1 outlines data collection and analysis methods, while Table 1 lays out demographic information. All study participants (mental health clinicians) worked in outpatient mental health services during the COVID-19 pandemic in 2020 and delivered at least some services via telehealth during this time. Clinicians participated in semi-structured one-to-one interviews guided by an interview schedule (see Appendix A). Semi-structured interviews were chosen as the data collection method as they provided sufficient scope for participants to voice their opinions around lockdown and telehealth service provision without having to follow a rigid structure applied during more structured interview approaches [27,28] and also without necessarily being influenced by other’s perspectives (a problem with focus group interview methods [29]). In turn, from a practical point of view, this data collection method meant that clinicians could participate in interviews during work time without the rest of the outpatient team’s work being interrupted.

To ensure adequate representation of Māori (NZ’s Indigenous population), all five Māori clinicians showing interest, including a kaumātua, were recruited. In keeping with Te Tiriti o Waitangi and Mātauranga Māori (Māori knowledge) research principles [30], all research participants could review and comment on their interview transcripts and interviews with Māori clinicians began and ended with a karakia (blessing). Steps were taken to treat Te Reo Māori words respectfully and appropriately within the analysis and reporting, such as confirming correct interpretation with Māori participants, and exploring Māori concepts, to gain a deeper understanding of Māori perspectives. The research team also undertook additional Tikanga Māori (Māori customary practices) research training and consulted with relevant regional advisory groups so that the project was underpinned by culturally safe protocols and concepts.

## 3. Results

Clinicians described experiences of the COVID-19 lockdown in NZ that intertwined considerable prior health service experience and training with the uncertainty of a new disease. These experiences were used to develop themes outlined in Figure 2.

### 3.1. Life in Lockdown

#### 3.1.1. Bracing for Impact

Few participants had experience with infectious disease outbreaks. Lack of COVID-19 information (including mode of transmission, incubation times, and rates of transmission) and uncertainty around the responsibilities of clinicians caused anxiety in the lead-up to the lockdown, due to fear of the unknown. Clinicians relied on international media coverage, as a result of limited local coverage of NZ’s COVID-19 response at a clinical level.


*I wasn’t sure how catastrophic the situation might develop… I was prepared to be really in an acute scenario similar to Italy, so no sleep, no contact with partner, to live in hospital environment, long working hours almost to the exhaustion.*
(P19)

Clinicians reported anxiety during the lockdown over NZ government requirements for the continued provision of healthcare in-person, with potentially infectious clients. Clinicians felt exposed to COVID-19, due to inadequate PPE (personal protective equipment) availability (masks and gowns), delayed training (for PPE use and minimising transmission), and a perceived lack of timely communication from employers.


*…there was no PPE gear, which I also didn’t know how to use.*
(P17)

Clinicians perceived a lack of guidance in setting up home workplaces, which worsened uncertainties. Teams reported inadequate office equipment or compatible devices to support clinicians working from home, leaving team leaders to problem-solve with limited resources.


*We only ever had four laptops across both teams [24 clinicians]… we automatically had to assume that our tangata motuhake [people special to the participant, in this case, other staff] had Zoom for starters, had equipment themselves in order to be able to do that.*
(P8)

Clinicians identified a lack of communication from health system decision-makers, which led to directives that were neither clinically useful nor fit for purpose. Clinicians felt this contributed to their anxiety and tarnished their lockdown work experience. Clinicians emphasised the need for decision makers to include clinicians in policy and planning discussions.


*Operations or management, the decree-ers, acknowledged that if they were to do it again, they’d probably do it differently and might actually listen more to what we were saying.*
(P8)

#### 3.1.2. Blurring Boundaries

Clinicians commonly expressed concern about notionally bringing clients and their distressing narratives into the clinician’s homes while consulting via telehealth. Working from home blurred boundaries between home and work, contributing to increased emotional burdens. To limit intrusion into personal spaces, clinicians consulting by videoconferencing removed personal items from view. Some clinicians with no home office separated themselves by using unconventional workspaces.


*Because bringing the work into your own bedroom, having some rather unpleasant conversations with some unpleasant people, there are those energies coming into the space of your own home, was a little bit tough… I started off thinking I could work in my garden shed, that was too uncomfortable. Then we tried to pitch a tent in the front garden.*
(P4)

Clinicians reported increased difficulty managing unpleasant situations when working-from-home. They were unable to maintain clear boundaries and had less collegial support to shield them from abuse. They described feeling apprehensive about receiving verbal abuse and some feared that clients, intending harm, could find where they lived based on their surroundings when videoconferencing. Clinicians felt this intrusion disrupted the neutrality of therapeutic relationships between client and clinician and made clinicians less secure.


*Some of my colleagues they were getting lots of verbal abuse from clients and felt attacked. That was a bit more confronting for them being at home… Whereas at work, you’ve got a bit more of a barrier up.*
(P6)

#### 3.1.3. Managing a Household

Clinicians undertaking in-person consultations were anxious about bringing COVID-19 back to their households. These individuals cordoned off areas for decontamination, showered on arriving home, washed work clothes separately, and created airlock systems to separate “the outer world and the inner world” (P19).

Clinicians with young families struggled to balance work and family commitments. Common reasons identified when delivering telehealth services included a lack of space to work effectively, difficulty managing interruptions from dependents and maintaining client confidentiality at home when other people were in the house, and insufficient resources for school-aged children (sharing laptops for school and work). Restricted access to day-care services and in-home support further forced solo-caregivers and partners of other essential workers to work from home while caring for children.


*Trying to share a space with them, and my husband was working-from-home… Space-wise, one was needing to work on the dining room table and I needed to work on the computer in the corner of the dining room, was tricky because whenever the phone rang, I had to run to another room and either shut myself in the living room for confidentiality or in my bedroom.*
(P15)

### 3.2. Collegial Support

#### 3.2.1. Double Bubble

Clinicians described splitting into sub-teams that alternated between telehealth consultations-from-home and outpatient clinics; this led to teams fragmenting and clinicians feeling isolated. This separation was compounded by concerns over ‘popping bubbles’ when clinicians from different teams shared offices or jointly consulted on cases. While the ‘bubble’ system was intended to minimise service disruption from clinician illness, fear of transmission between teams established ‘care silos’, and smaller teams struggled to provide services with available staff.


*We’re a small enough team as it is and having to split, we’re so used to working together as a team, and being a small team, one person that’s away can throw the rest of the team’s work, especially if there are issues with clients.*
(P33)

Before the pandemic clinicians had close team relationships and could support one another. Lockdown saw clinicians struggling to identify when colleagues required support, as when teleworking, they were unable to see social cues and respond naturally. Clinicians tried to address this issue by organising regular informal online well-being sessions and collegial support groups. However, many struggled with using technology or could not attend these sessions.


*You kind of feel lost without having that instant team interaction like someone sitting next to you or you’d just be able to walk down the hallway and ask for help… so you do feel isolated.*
(P13)

Clinicians suggested that teams often sequestered their doctors, as the doctor’s absence would limit the service’s functioning. Clinicians, not participating in in-person reviews, described feeling guilty about not sharing team burdens and adding to workloads.


*There was some lack of equity really because my colleagues although they were working week on and week off, they were obliged to go out into the community and have potentially infectious contact with people on a regular basis every day and I didn’t have to do that.*
(P14)

#### 3.2.2. Corridor Consultations

Before the lockdown, clinicians described consulting informally with colleagues on clinical issues. These ‘corridor consults’ were opportunistic and allowed for a timely discussion on cases. Clinicians found that physical separation and teleworking reduced their ability to have corridor consults, which led to feeling less supported by colleagues, and clinicians postponing time-sensitive discussions until meetings could be organised.


*Hallway discussions are actually quite important. Many a time I’ve ran a discussion that’s turned into more of… a mini-MDT [multidisciplinary team], organically happens… those office discussions that are often valuable which we didn’t have with COVID.*
(P3)

In contrast, some clinicians identified telehealth as a new way to access opportunistic consultations. Clinicians familiar with using telehealth had fewer difficulties arranging discussions with colleagues at short notice. They also found that they were more accessible to their colleagues. Furthermore, clinicians were able to share resources, such as research articles to support discussions.


*To be accessible and that you can access colleagues for professional meetings in an easier way… as a supportive communication within the professional field, definitely a must. Parts I like are screen sharing, for a lot of discussions, it’s becoming an absolutely vital tool. Talking about data, talking about information sharing, developing ideas.*
(P19)

#### 3.2.3. Supervision

Doctors, nurses, and psychologists have established peer groups and individual supervision. Several clinicians described the value of supervision in managing their psychologically challenging work. However, clinicians identified a need to develop formal supervision programmes for other health professionals (social workers, occupational therapists, team leaders, case managers, and cultural liaison workers).


*If there was some sort of buddy system where people are encouraged to match up with somebody or have a smaller group and do that, that would be really cool. Yeah, maybe EAP [employee assistance programme] but a bit more casual.*
(P21)

Clinicians accessing supervision found that online tools increased their ability to attend supervision. Videoconferencing allowed clinicians greater freedom to contact supervisors and remove geographical barriers. These clinicians described continuing videoconferencing supervision after the lockdown due to its improved convenience.


*During the lockdown, you were able to contact me on other days rather than just when we were seeing each other face to face, so I think in some ways it was an advantage. In some ways, it improved access because the medium of communication was technology and there was less of a barrier.*
(P17)

Supervising psychiatrists and clinical psychologists felt that videoconferencing was an adequate alternative to observe trainees conducting consultations and provide them with psychological support.


*We got quite a bit of information from that OCE (observed clinical examination) about that [client]… I think as a consultant, because of the on-call situation, I’m used to being on the phone… Zoom was kind of an added bonus because we could see each other.*
(P17)

### 3.3. Maintaining Wellbeing

#### 3.3.1. Balancing Risks

Clinicians identified a need for limited in-person contact with higher-risk clients, because of the severity of the client’s mental illness, and the requirements to monitor physical comorbidities affected by care decisions. Clinicians recognised the client’s unsafe home environments, reluctance to use face masks, and difficulty self-identifying COVID-19 symptoms as compounding risks. Furthermore, clients with severe mental illness requiring admission to inpatient units also may have required restraint, raising the risk of virus transmission.


*I did go in and see somebody face-to-face at the hospital, they weren’t socially distancing, they weren’t wearing masks. And at that point, we thought every surface had COVID on it.*
(P12)

Clinicians directly balanced the reality of delivering in-person services against COVID-19 exposure fears. They wore full PPE for each in-person consultation. This was time-consuming, resource-intensive (requiring new PPE for each contact), and limited client interactions.


*I was kind of worried that we weren’t going to be able to keep safe and make adequate assessments… I think masking, when you’re seeing people face-to-face, is difficult, because you’re not showing your expression and they’re not showing their expression if they have a mask on.*
(P12)

#### 3.3.2. Healthy Practices

Leading up to the lockdown, clinicians described not prioritising their mental well-being, due to high workloads and stress. During the lockdown, many clinicians felt their workloads reduced, allowing them to recover from accumulated stresses. Clinicians suggested that working in mental health helped them implement practices to maintain their well-being and manage lockdowns. Few clinicians reported deteriorations in physical well-being (such as weight changes) caused by working at home despite extensive screentime. Instead, clinicians adopted a wide variety of management strategies (including working on culinary skills and improving diet and exercise), with most adapting routines due to lockdown restrictions.


*I was able to get out more, so I thought that that was nice to have that mix of work and getting some fresh air because you could still go for a walk… It felt like it kept my sanity a bit more intact. Because, for me, I lose focus quite easily, so the stopping and starting and coming back and focusing for shorter periods of time is pretty effective.*
(P28)

However, clinicians also explained that when working from home, there were no scheduled disruptions bringing them respite. They described ‘Zoom fatigue’ and eye strain caused by a lack of familiarity with telehealth, constant shifting attention during video conferencing, and extended screentime. This shifting of attention occurred both in videoconferencing with multiple parties each of whom the clinician was assessing or managing interactions, and when trying to gauge clients’ physical signals while concentrating on facial movements during telehealth consultations.


*We were really surprised at how fatigued we got doing Zoom. It was really, really tiring… because you are focused on a small screen and also it’s very intense. You’re having to be on the whole time because you’re in front of a camera so you can’t be doing things… I think it might be that having to concentrate a lot more and having to listen to multiple people.*
(P1)

Clinicians often reported decreased job satisfaction during lockdown because telehealth consultations restricted the quality of interpersonal client relationships. Clinicians felt they were ticking off checklists and were unable to use all their skills to engage clients. For example, clinicians could not take clients for walks or support clients at appointments. Clinicians explained that these activities usually created opportunities for therapeutic interactions that were unavailable in telehealth-only services. Some clinicians suggested that personal contact was a core reason for working in mental health and removing in-person contact led to feeling ineffective and devalued.


*It’s definitely much nicer being back and seeing clients face-to-face. That’s why I’m in this job. Not seeing people face to face certainly does take something away. I guess that’s why we do the job that we do because that’s what we enjoy.*
(P26)

#### 3.3.3. Māori Cultural Safety

Kaimahi (Māori clinicians) faced additional challenges during the lockdown because of (1) frustration caused by the temporary relocation of the Māori mental health service into a joint space and (2) separation from local communities and clinicians when working from home. These clinicians described negative impacts on their mauri (lifeforce) and wairua (spirit).


*It was that knee-jerk reaction… but especially here as a kaupapa (Māori knowledge and values) service, it was really hard… on our energy and our mauri and our wairua.*
(P8)

Kaimahi identified that kaumātua (respected elder) provided psychological and cultural support, similar to the informal supervision other clinicians received. Cultural support offered a sense of safety, and reduced anxiety and stress.


*When things are down it’s important for myself and our kaumātua to jump in and support our kaimahi because we know in this mahi (work), burnout is so intense, it’s so high. It’s again, replenishing our mauri and reinforcing that.*
(P11)

Kaimahi described isolation from clients and communities during the lockdown, due to their reduced in-person care. They expressed that Māori value physical connections and they returned to in-person service provision after the lockdown to regain this connection.


*There’s no aroha (love)… we hug people. We let people cry, we cry with them. It’s not the same connection, there’s no connection over the phone that there is when you see people.*
(P10)

Kaimahi practiced cultural-specific activities, including karakia (blessing) and waiata (song) that helped maintain well-being and team bonding. A tikanga (Māori customary practices) programme also continued via videoconferencing over lockdown to further clinician understanding of cultural practices. This gave clinicians a sense of normality and cohesion while physically separated. For Kaimahi, maintaining connections with their heritage created an appreciation that their values were respected, and for non-Māori, it provided insight into Māori traditions.


*Every morning [we started] with a karakia together as a team. Then I introduced what we called a mauri session… Mauri is one of the components of wellness within our workspace… Tikanga [customary practice] wasn’t just about looking at a Māori paradigm, it was about [implementing] the tikanga here for the whare [house].*
(P11)

## 4. Discussion

This research is one of the first to investigate the effects of the COVID-19 pandemic and associated lockdowns on mental health clinician experiences in NZ. It has the unique advantage of covering multiple regions, outpatient teams, subspecialties, and has Māori representation. This means that the 33 clinicians included in our research presented varied perspectives on service delivery over COVID-19 lockdown.

COVID-19 lockdown placed additional pressures and uncertainties on clinicians, a finding supported by Sibeoni, et al. [31], and World Health Organization [32]. These uncertainties do not disappear overnight, and require systemwide changes to support clinicians [33]. Our research identifies factors contributing to their anxiety, including systems factors, individual circumstances and the need for cultural safety. Building on recommendation from Dwyer [34] and Callaly and Minas [35], system-level changes to ameliorate these pressures could include addressing (1) disparate access to basic resources (PPE, desks, chairs and suitable work computers), (2) clinician inclusion in service planning, and (3) improving communication between senior management and health professionals. Future service continuity planning should consider how best to reorganise sub-teams so that specialist clinicians can operate across a service rather than solely with one team, for example, through establishing monitored disaster relief locum pools.

Challenges to wellbeing, the blurring of boundaries, and the need for greater support while working-from-home are consistent with findings arising from studies of the general population [5,13]. Although part of a good therapeutic relationship requires building trust and shared understandings [36], working-from-home can blur personal and professional positions with greater sharing of information about home environments [13,37]. Working-from-home may also redefine clinician regular work patterns, particularly within rapidly changing work contexts. Steps employers can take to support clinicians could include developing best practice guidelines on creating safe spaces and boundaries to conduct work at home. These guidelines could focus on ways to maintain privacy, manage household members, and develop a sustainable work routine.

With time and experience, clinicians were able to use videoconferencing for supervision and team meetings but found it difficult to provide collegial support. As NZ moves away from acute COVID-19 management, the health system should further consider the pandemic’s impact on clinician retention and the need to support those now working in hybrid (telehealth and in-person) arrangements. A survey of UK doctors indicated 21% intended to resign due to burnout and unmanageable workload related to COVID-19 [38], with similar findings amongst New Zealand clinicians [39]. Browne, et al. [40] provide recommendations, including the use of counselling to support clinicians’ mental health and reduce burnout. Our research develops these ideas from an NZ perspective and proposes the need for informal collegial support and supervision while working from home. This should include protected time for clinicians to provide peer support, mentoring, and formal supervision. Inequitable access to supervision for occupational therapists, social workers, and case managers should be addressed as part of these protocols, in line with Snowdon, et al. [41]. For Māori clinicians, Indigenous researchers recommend the need for wraparound support for clinicians to improve familiarity with mental health services and telehealth technology [42]. Our research further emphasises the need for cultural safety through consulting with Māori stakeholders on how best to apply telehealth as part of usual care. Furthermore, fostering relationships with local communities could support ongoing connection with (Māori and non-Māori) clients when required to work from a distance.

Like any qualitative research, findings generalisability is not possible, nor is it the intention. However, this research is timely, as NZ launches a Royal Commission of Inquiry into the NZ COVID-19 response, announced on 6 December 2022 [43]. It is important to bring clinician experience into these discussions, due to the impact their experiences have on service provision, staff retention, and service planning. Further research into clinician well-being following the peak of COVID-19 is required, particularly given the increasing mental health and health system burdens caused by delayed care [44,45]. Such work could focus on evaluating support systems introduced for clinicians, reviewing the impact of COVID-19 lockdown policies on service provision, and evaluating lessons learnt by clinicians over later lockdowns. Relevant recommendations for mental health outpatient teams and health system management are laid out in Table 2.

## 5. Conclusions

COVID-19 lockdown had diverse impacts on NZ mental health clinicians and led to blurred boundaries and changes in workplace practices. It has also presented a unique opportunity to identify learnings for business continuity and process improvements relevant to mental health service provision. Clinicians highlighted key processes related to collegial support, balancing risk, and ensuring well-being that enabled them to manage lockdown. Lessons derived from the pandemic emphasise the need for health system preparedness to manage the long-term ramifications of changing workplace practices (hybrid working models, telehealth consultations, and increasing service demand). Clinicians require a voice to ensure that their well-being is considered during policymaking and service planning. Failure to do so means that clinicians will continue to face high-stress and low-retention environments without necessary support, and that the health system has not learned from this pandemic when planning for the next.

## Figures and Tables

**Figure 1 ijerph-20-04791-f001:**
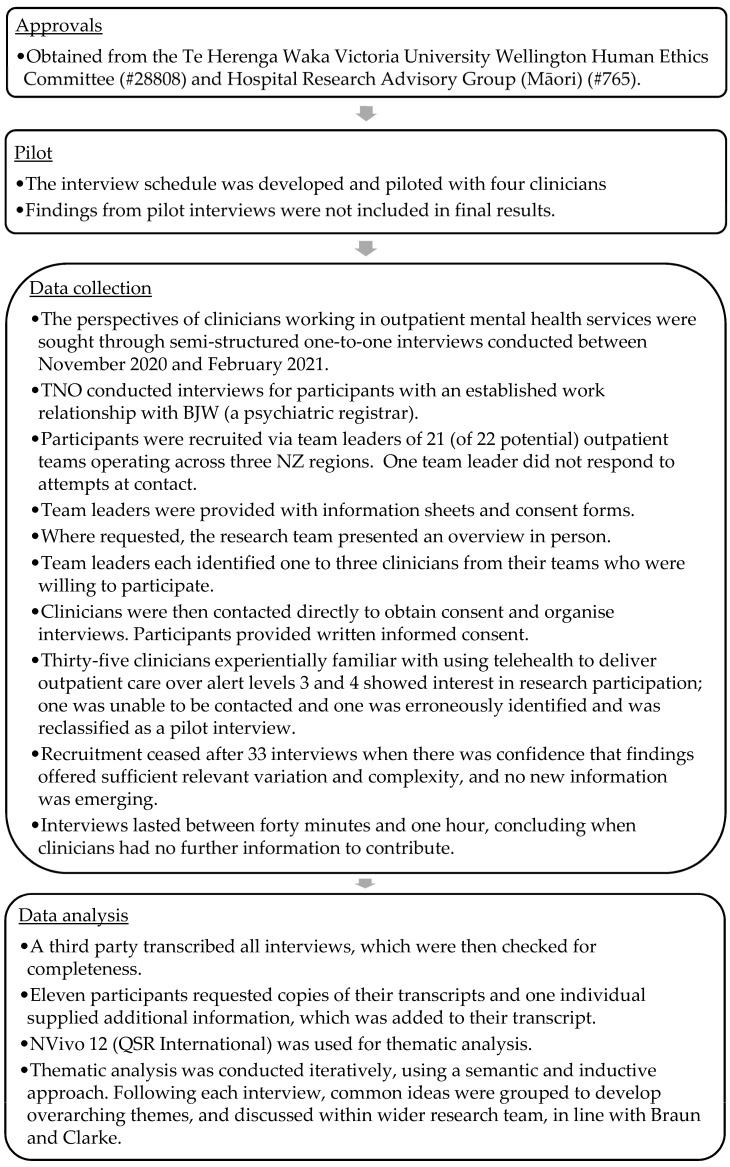
Data collection and analysis.

**Figure 2 ijerph-20-04791-f002:**
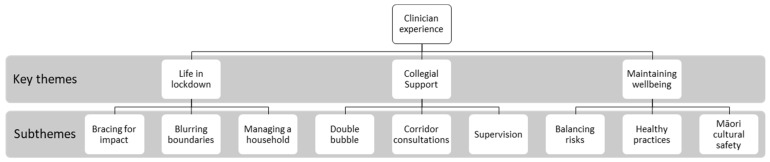
Common themes.

**Table 1 ijerph-20-04791-t001:** Participant demographics.

Role		Ethnicity		Age Range		Gender	
Doctor	7	Māori	5	25–35 years	6	Male	10
Nurse	6	NZ European	18	35–44 years	6	Female	23
Clinical psychologist	5	Other European	4	45–54 years	10		
Team leader	5	Pacific	2	55–65 years	6		
Social worker	4	Other	4	65+ years	5		
Occupational therapist	2						
Kaumātua	1						
Cultural therapist	1						
Psychotherapist	1						
Case manager	1						
Total	33	Total	33	Total	33	Total	33

**Table 2 ijerph-20-04791-t002:** Recommendations for outpatient teams and health system management.

Outpatient Teams	Health System Management
Address disparate access to basic resources (PPE, desks, chairs, and suitable work computers).	Include clinician stakeholders in service planning and policy making.
Develop best practice guidelines for creating safe spaces and boundaries to conduct work at home.	Improve communication between senior management and health professionals.
Foster relationships with local communities to support ongoing connection with (Māori and non-Māori) clients.	Establish monitored disaster relief locum pools.
Develop collegial support and supervision networks.	Improve cultural safety through consulting with Māori stakeholders.
Include protected time for clinicians to provide peer support, mentoring, and formal supervision.	Implement and review policies for monitoring clinician well-being during pandemics or as part of emergency management.
	Create supervision protocols for occupational therapists, social workers, and case managers.

## Data Availability

Not applicable.

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
