# Peer review of "Experiences of the COVID-19 Lockdown and Telehealth in Aotearoa New Zealand: Lessons and Insights from Mental Health Clinicians"

_ijerph, 2023, doi:10.3390/ijerph20064791_

Round 1
Reviewer 1 Report
In some places of the methodology and results it is not clear whether they are talking about telemedicine or not. The title need to be aligned with the body.
Author Response
Dear Reviewer,
Please see attached letter in response to your feedback.
Manny thanks for engaging with our manuscript.
Benjamin Werkmeister (on behalf of the authorship team).

Reviewer 2 Report
Overall, this is an interesting study and a well-written paper that I enjoyed reading. I offer a couple of minor comments:
Lines 66-74 should be removed – they are instructions to the authors.
Reference problems in the document when trying to refer to other sections.
Line 198, ‘mini-MDT’ – spell out in brackets what an MDT is.
Line 259 suggests that there were not physical deteriorations reported due to screen time, but line 269 suggests that eye strain was reported. This needs to be more clearly presented so that it does not seem to conflict.
Author Response

(The authors gave the same response as above.)

Reviewer 3 Report
- I appreciate this article and the timing of not only considering the impact of these mental health professionals while in the midst of the working conditions of the pandemic, but also the forward-thinking approach to improve practices for future pandemics. The perspectives from a sampling of the mental health clinicians in NZ, including Maori, is important information for NZ and the future considerations.
- General concept comments
Article: I would like to see more information about the chosen qualitative methodology, including the why and what other methods were considered and ruled out. Just because a study is qualitative in nature does not mean we forego the consideration of the possible methods to determine which is most appropriate.
Review: I found the introduction to be lacking in a number of ways. First, there was no transition to explain why you sifted from the overall focus of the population of NZ to the specific mental health clinicians (line 42-43). Given you inclusion of Maori clinicians (which is important for NZ!), there needs to be information in the introduction regarding the Indigenous population. Given this is an international journal, there appear to be a number of assumptions made from the NZ lens that do not consider the broader audience as well, including the assumption of knowledge of the Maoiri. T. - Specific comments Additional examples for the assumptions: in the data collection, psychiatric registrar is a term used. I do not know what this means, as this is not a term used in my country. I am also confused about the inclusion of some of the participants (Table 1), as doctors, nurses, and occupational therapists are not considered mental health clinicians in my country, unless they are specific specialists (in which case, their mental health specialty would be what is relevant for this study). Thus, I do not know that their interview data is actually relevant to this study, as it is the mental health clinicians you indicate focusing on. If you intend to keep these other professionals in this study, then you would need to expand the paper to go beyond mental health clinicians. I am also not sure why "team leader" is relevant--it would again, be their mental health title I would want to see.
- Although I know it can happen when editing ones work, writers should not have errors in references (see, for example, lines 78-79; 88-89) nor should material from the original template be left in the draft to be reviewed (see lines 66-74).
- Line 329. Please be specific again that you are referring to NZ specifically.
- Given what is mentioned about the Royal Commission (line 373), I would have liked to see a list of key recommendations based on this study, perhaps in bullet form.
Author Response

(The authors gave the same response as above.)
